# Artificial diet alters activity and rest patterns in the olive fruit fly

**Anastasia M. Terzidou[1], Dimitrios S. Koveos[1], Nikos T. Papadopoulos[2], James R. Carey[3,4], Nikos A. Kouloussis[1]***

**1** Laboratory of Applied Zoology and Parasitology, School of Agriculture, Aristotle University of Thessaloniki, Thessaloniki, Greece, **2** Laboratory of Entomology and Agricultural Zoology, Department of Agriculture, Crop Production and Rural Environment, University of Thessaly, N. Ionia Magnisia, Greece, **3** Department of Entomology, University of California, Davis, Davis, CA, United States of America, **4** Center on the Economics and Demography of Aging, University of California, Berkeley, Berkeley, CA, United States of America

* nikoul@agro.auth.gr

**Data Availability Statement:** https://doi.org/10.5281/zenodo.7221901.

**Funding:** Research funded in part by a grant from the Center for the Economics and Demography of

## Abstract

Olive fruit flies, *Bactrocera oleae* (Diptera: Tephritidae) reared in the laboratory on an artificial diet are essential for the genetic control techniques against this pest. However, the colony's laboratory adaptation can affect the quality of the reared flies. We used the Locomotor Activity Monitor to track the activity and rest patterns of adult olive fruit flies reared as immatures in olives (F2-F3 generation) and in artificial diet (>300 generations). Counts of beam breaks caused by the adult fly activity were used as an estimation of its locomotor activity levels during the light and dark period. Bouts of inactivity with duration longer than five minutes were considered a rest episode. Locomotor activity and rest parameters were found to be dependent on sex, mating status and rearing history. In virgin flies reared on olives, males were more active than females and increased their locomotor activity towards the end of the light period. Mating decreased the locomotor activity levels of males, but not of female olive-reared flies. Laboratory flies reared on artificial diet had lower locomotor activity levels during the light period and more rest episodes of shorter duration during the dark period compared to flies reared on olives. We describe the diurnal locomotor activity patterns of *B. oleae* adults reared on olive fruit and on artificial diet. We discuss how locomotor activity and rest pattern differences may affect the laboratory flies' ability to compete with wild males in the field.

## 1. Introduction

The olive fruit fly has been the most important pest of olives in Mediterranean countries for at least 2000 years [1] and in 1998 it was first detected in California [2]. Its distribution now covers the Mediterranean basin, South and Central Africa, south-west Asia, Pakistan, California and Mexico [3, 4]. Because of the significant economic damage caused by *B. oleae* and the intense use of insecticides to manage it, Sterile Insect Technique (SIT), as well as a self-limiting olive fly technology have been proposed to control this pest [5, 6]. Both methods require the mass-release of quality laboratory male olive fruit flies reared on artificial diet, that will be able to survive, search, find, and mate effectively with the wild population. Selection during

Aging, UC Berkeley (recipient: JRC) (grant number: NIH 2P30AG012839) which covered the research equipment and publication fee and in part by Greece and the European Union (European Social Fund-ESF) through the Operational Program «Human Resources Development, Education and Lifelong Learning» in the context of the project "Strengthening Human Resources Research Potential via Doctorate Research" (MIS-5000432), implemented by the State Scholarships Foundation (received by Anastasia M Terzidou).

**Competing interests:** The authors have declared that no competing interests exist.

colonization and mass-rearing normally changes the biology and behavior of an insect species. A common problem in mass-reared insects is the loss of irritability, because crowded rearing conditions select for individuals that ignore movements in their surroundings [7]. Mass-rearing conditions reduce the flight capacity of *Amyelois transitella* [8], and the overall activity of *Bactrocera tryoni* compared to wild population [9], while inbreeding in *Drosophila melanogaster* decreases locomotor activity and changes daily activity patterns [10]. It has been shown that in *Bactrocera dorsalis*, the time of mating can be altered in insects that have been adapted in laboratory rearing conditions, even after a period of just a year [11].

Sleep and rest states are present in all animals and many studies have shown that they exist even in arthropods and nematodes [12]. Sleep is necessary for an animal's survival and health, including replenishment of energy stores, removal of harmful by-products, and maintenance of neural plasticity [13]. There is also evidence that night sleep disruption, including sleep deprivation (total sleep loss) and sleep restriction (partial sleep loss), caused by a variety of biotic and abiotic factors in nature can cause changes in insect daily activity patterns with consequences in behavioral performance during active periods [14].

Daily rhythms of activity and rest can be recorded by placing individuals in glass tubes and monitoring the movements using infrared beam-based activity monitors like Locomotor Activity Monitor-LAM by Trikinetics (https://trikinetics.com/) [15]. One limitation of the LAM tracking system is that it cannot differentiate between inactivity and sleep [16]. For bumblebees, bouts of inactivity lasting more than 5 min during the night were visually associated with posture indicating sleep, but during the day, inactivity periods could not reliably be associated with sleep [17]. In Drosophila, which has been used extensively as a model organism for sleep studies [18, 19], the 5 min threshold of inactivity for defining a sleep state is widely accepted and the duration of its sleep episodes during the night indicate how well a fly can stay asleep [20]. In Drosophila and mammals, the sleep state should be accompanied by changes in arousal threshold [21] among other parameters, like body posture and preferred location.

The aim of this study was to record the diurnal patterns of locomotion of olive fruit flies, as a detailed tracking of their activity with the use of LAM. We focused on reproductively mature wild flies (F2-F3 generation reared on olive fruit) and how their diurnal patterns of locomotion change according to sex and mating status. We also studied the effect of laboratory rearing on artificial diet to the locomotor activity and rest patterns of reproductively mature virgin male flies. We hypothesized that wild flies reared on olives would have higher locomotor activity levels than laboratory flies reared on artificial diet, as laboratory mass rearing is known to further decrease the tendency of flies to move [9].

## 2. Materials and methods

### 2.1. Insect rearing

The wild olive fruit fly colony was established with flies that emerged from infested olives, field collected in late September from non-chemically treated olive trees in the area of Thessaloniki. No permission field access or fruit collection was needed, as trees were not part of commercial grove, but either abandoned or planted for ornamental purposes. Emerged adults were maintained in wooden cages (30 x 30 x 30 cm), approximately 180–200 flies per cage, under laboratory conditions (24±1.5°C, RH 40±5%, L:D 14:10). During the photophase, light was provided by typical neon tubes (Philips18W/865), at light intensity of approximately 1000 lux near the cages (measured with Elvos Luxmeter LM1010). Food provision was in the form of a liquid diet consisting of sugar, yeast hydrolysate enzymatic (Chembiotin) and water (ratio 4:1:5). Flies were allowed to oviposit in olives. Water was provided with a soaked cotton wick extruding from a small water container. Flies of this wild population completed their larval stages in

olives for 2–3 generations in our laboratory and were used in our experiments (hereafter referred to as W flies).

We used laboratory adapted olive fruit flies reared on artificial diet during their larval stages (hereafter referred to as AR flies) from the colony maintained in our laboratory for more than 20 generations. The colony was established from the "Democritus strain", which was developed in the Democritus Nuclear Research Center, Athens, Greece and had already been reared for more than 300 generations. Adult flies were kept in wooden cages (30 x 30 x 30 cm), and each cage contained about 200 individuals. Adult food was the same as for W flies and given in the form of a liquid diet consisting of sugar, yeast hydrolysate enzymatic (Chembiotin) and water (ratio 4:1:5) (no antibiotic was added). Egg yolk powder was added ad libitum as extra protein source for the colony AR flies. They were allowed to oviposit on beeswax domes (diameter = 2 cm) and eggs were collected every two days with a fine brush and washed with propionic acid solution (0.3%). The collected eggs were then placed directly on the larval diet inside a Petri dish (94 x 16 mm). Larval artificial diet consisted of 550 ml of tap water, olive oil (20 ml), Tween 80 by Sigma-Aldrich (7.5 ml), potassium sorbate by Sigma-Aldrich (0.5 g), nipagin (Methyl-4-hydroxybenzoate) by Sigma-Aldrich (2 g), crystalline sugar (20 g), brewer's yeast (MP Biomedicals, LLC) (75 g), soy hydrolysate Peptone from Glycine max (Soybean) Type IV powder by Sigma-Aldrich (30 g), hydrochloric acid 2N (30 ml), and cellulose powder as bulking medium (275 g) as described in Tsitsipis et al [22]. The diet was in granulate form after the addition of cellulose and was kept moist to stimulate last stage larvae to exit the diet which were then collected by sieving the sand on which the Petri dish was placed.

Newly emerged W and AR flies were separated by sex in the first 24h of their emergence, kept in plexiglass cages (15 x 15 x 15 cm) in groups of 20 and under the same conditions (T: 24 ±1.5°C, RH: 40±5%, L:D 14:10) and fed with the same liquid diet consisting of sugar, yeast hydrolysate enzymatic and water (ratio 4:1:5).

Male and female W flies of the same cohort were kept together after adult emergence in colony wooden cages (about 70–90 flies per cage) under the same conditions and diet as above. They were considered mated by the time they were used for the bioassays (12–13 days old at the beginning of bioassay).

Laboratory adaptation results in more rapid sexual maturity rates for AR flies [23, 24]. Sexual maturity begins on the 8th day of age of W flies in both sexes [25], whereas in AR males it begins on the 2nd day and in AR females on the 3rd day of age. W flies used were 12–13 days old at the beginning of the bioassay, while AR flies were 6–7 days old, considering the AR flies' shorter longevity [26] and that AR flies for SIT purposes are released at a young age [27].

## 2.2. Locomotor activity

We recorded the locomotor activity patterns of adult olive fruit flies using the Locomotor Activity Monitor- LAM25 system (Trikinetics Inc, Waltham, MA, USA). In this system, flies were individually kept in 32 glass tubes with 25 mm diameter and 125 mm length. A vinyl plastic stopper (CAP25-BLK- Vinyl Tube Cap-Trikinetics) was adjusted inside on the one end of the tube, which was an agar-based gel diet with sugar, yeast hydrolysate, agar, nipagin and water (4:1:0.2:0.1:20), for food and water provision [28]. The other end of the tube was covered with a piece of organdie to allow ventilation. The tubes were maintained in a climatic room under a photoperiod of 14:10 L:D. The light period was from 07:00 to 21:00 (hereafter referred to as LP), and dark period was from 21:00 to 07:00 (hereafter referred to as DP). Activity at each tube was measured every minute as counts of infrared light beams crossed.

Three LAM devices were used simultaneously for the W flies bioassay. Thirty-two W virgin males and thirty-two W virgin females were maintained in two LAMs. Sixteen W mated males

and sixteen W mated females were maintained in the third LAM. They were monitored for 5 consecutive days. After the completion of this bioassay, AR virgin flies (sixteen males and sixteen females) were maintained in one LAM device and monitored for 5 consecutive days. The number of mated W and virgin AR flies used per treatment, although small, is the same as in the work of Chiu et al. [29] where n = 16 *Anastrepha ludens* flies were used per treatment.

### 2.3 Visual observation of resting flies

We visually observed resting flies during LP and DP for their preferred body posture and resting location in the glass tube. Only wild virgin olive fruit flies were used for this bioassay. Observations during DP were done under red light. Ten female and ten male W virgin flies aged 8–9 days old were maintained in the same glass tubes and with the same food/water provision as previously described for the LAM bioassays. We used a TrueLUX (HC26900) macro camera lens attached to a smartphone camera for photo and video capture of the flies.

### 2.4. Data analysis

The LAM devices were set to record the sum of movements each fly performed every minute and export the data in monitor files as the number of counts for each tube. The raw monitor data were processed in the DAM FileScan software and activity data collected in 1-minute intervals (1-minute bins) were compressed and converted to 30-minute intervals (30-minute bins) for plotting purposes. Activity/rest analysis was performed using an in-house MATLAB program called Sleep and Circadian Analysis MATLAB Program (SCAMP) [30]. Activity levels during the LP and DP were the total counts for each fly during the 14 h period of lights on and 10 h period of lights off respectively. Since we could not associate the inactivity of flies as tracked by the Locomotor Activity Monitor with the sleep state, we refer to bouts of inactivity longer that 5 min as rest episodes.

Activity levels, number and duration of rest/sleep episodes were calculated as the average of 5 consecutive days of monitoring after excluding the flies that died during the bioassay. Sample size was *n* = 32 W virgin flies of each sex, *n* = 16 AR virgin flies of each sex, and *n* = 12 male and *n* = 14 female W mated flies. For all the comparisons, a 2-tailed *t*-test (was) performed (level of significance α = 0.05) with the statistical software package JMP 14.1.0 [31].

## 3. Results

### 3.1. Locomotor activity

**3.1.1. Locomotor activity of W flies.** The pattern of locomotor activity of W virgin males and females during 24-h in 30 min bins is shown in Fig 1A. The mean locomotor activity level (±SE) of W virgin males during the LP was 2031.3 ± 132.0 counts and that of W virgin females was 1394.2 ± 86.6 counts. They differed significantly (2-tailed *t*-test = 4.035, *df* = 54, *P* = .0002). However, during the DP, the mean locomotor activity level (±SE) of W virgin males and W virgin females was 257.2 ± 17.8 counts and 246.6 ± 6.3 counts respectively and did not differ significantly (2-tailed *t*-test = .438, *df* = 62, *P* = .662). High levels of locomotor activity during the DP were recorded at the time of lights off and for the next hour after the transition to scotophase.

The pattern of locomotor activity of W mated males and females during 24-h in 30 min bins is shown in Fig 1B. The mean locomotor activity level (±SE) of W mated males and W mated females during the LP was 1153.1 ± 174.0 counts and 1081.9 ± 181.5 counts respectively and did not differ significantly (2-tailed *t*-test = .283, *df* = 24, *P* = .779). The mean locomotor activity level (±SE) of W mated males and W mated females during the DP was 208.2 ± 30.5

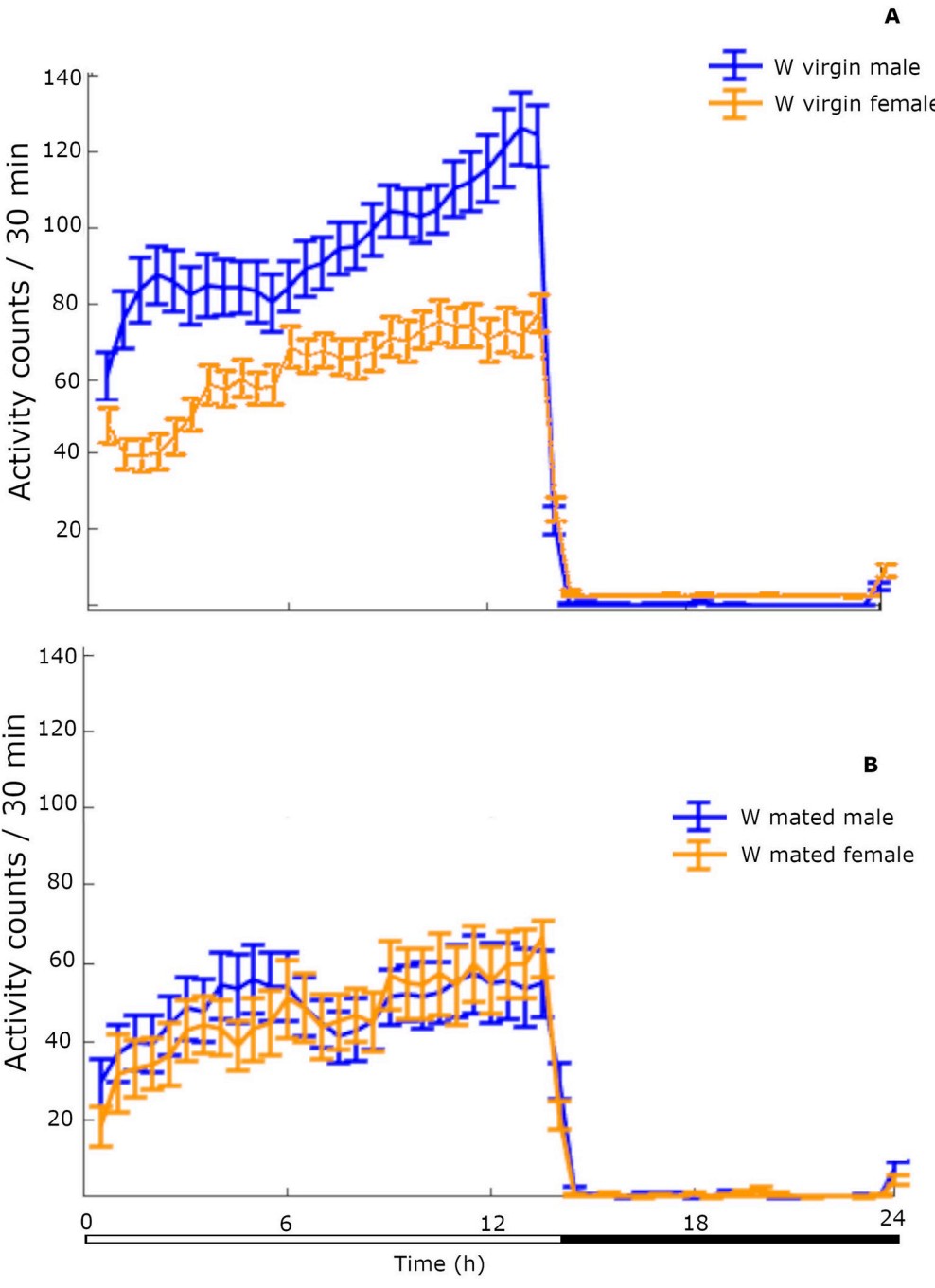

**Fig 1. Pattern of locomotor activity of W virgin and mated flies of both sexes across 24-h.** Locomotor activity levels (±SE) of W virgin male and female flies (A) and W mated male and female flies (B) ($n$ = 32 W virgin flies of each sex, $n$ = 12 W mated males, $n$ = 14 W mated females averaged across 5 days of monitoring).

counts and 219.0 ± 26.3 counts respectively and did not differ significantly (2-tailed $t$-test = -0.266, $df$ = 23, $P$ = .792).

Mating affects the locomotor activity of males but not of females during LP. The mean locomotor activity was significantly higher in virgin W males compared to mated ones during LP (2-tailed $t$-test = 4.020, $df$ = 24, $P$ = .0005) but not during DP (2-tailed $t$-test = 1.383, $df$ = 19,

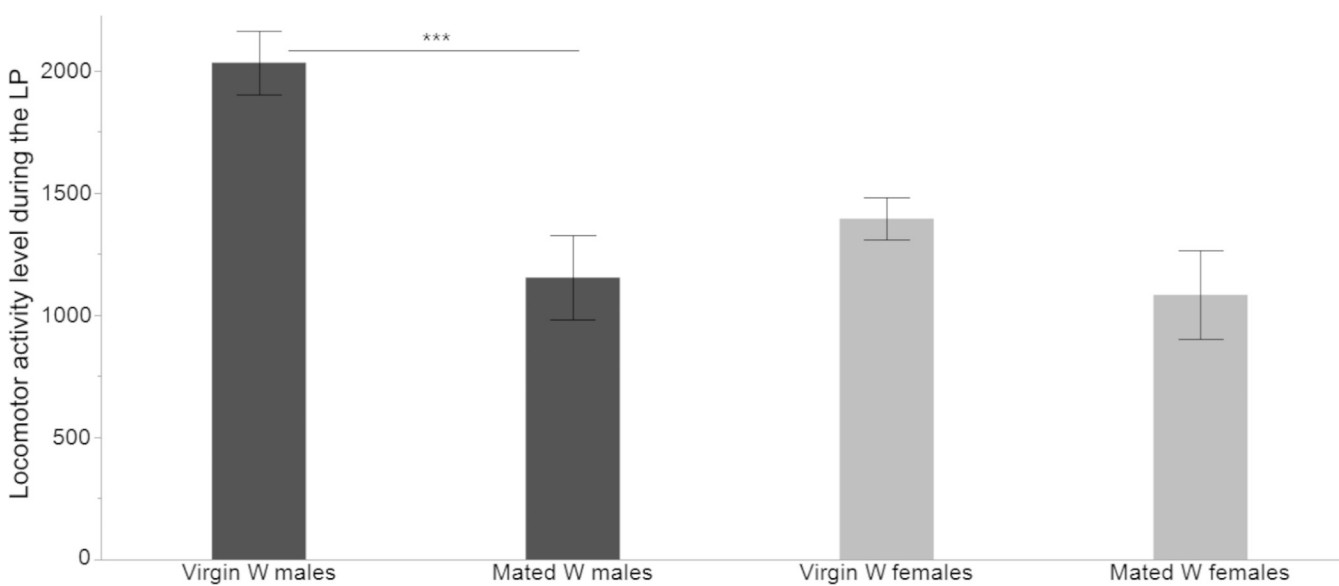

**Fig 2. Comparison of locomotor activity levels of W virgin and mated flies during the LP.** Mean (±SE) level of locomotor activity of W virgin and mated flies of both sexes as total counts of activity during the LP ($n$ = 32 W virgin flies of each sex, $n$ = 12 W mated males and $n$ = 14 W mated females, averaged across 5 days of monitoring) (*** $P$ < .001).

$P$ = .182). However, virgin and mated W females did not differ significantly in their mean locomotor activity during LP (2-tailed $t$-test = 1.553, $df$ = 19, $P$ = .136) or DP (2-tailed $t$-test = .889, $df$ = 23, $P$ = .382). The differences between virgin and mated flies in their locomotor activity leves during the LP only are shown in Fig 2.

**3.1.2. Locomotor activity of AR flies.** The pattern of locomotor activity of AR virgin males and females in 30-min bins across 24 h is shown in Fig 3. The mean locomotor activity level (±SE) of AR virgin males and AR virgin females during the LP was 363.0 ± 63.2 counts and 324.3 ± 45.6 counts respectively. Contrary to the W flies, in AR flies there was no statistical

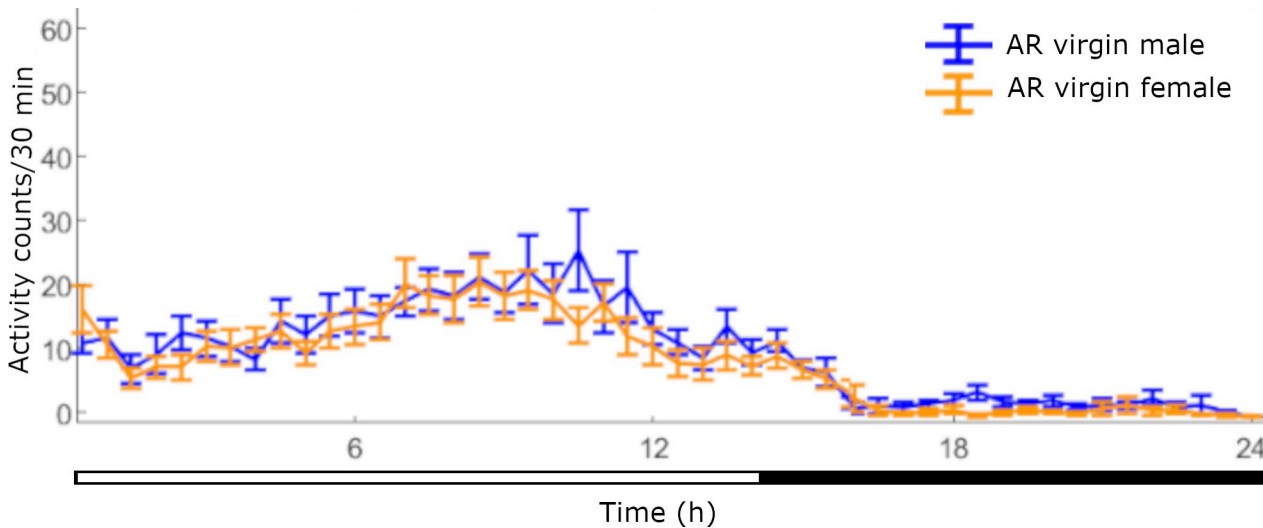

**Fig 3. Pattern of locomotor activity of AR virgin flies of both sexes across 24-h.** Locomotor activity levels (±SE) of AR virgin male and female flies ($n$ = 16 flies of each sex, averaged across 5 days of monitoring).

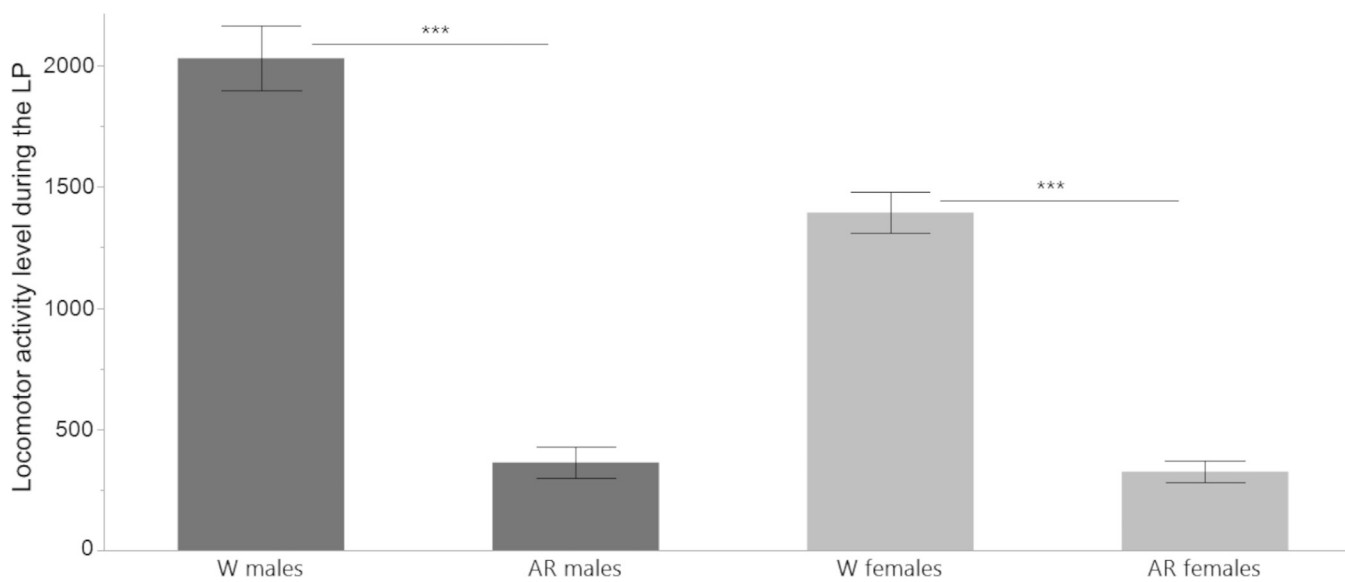

**Fig 4. Comparison of locomotor activity levels between W and AR virgin flies during the LP.** Mean (±SE) levels of locomotor activity of W virgin and AR virgin flies or both sexes as total counts of activity during the LP (*n* = 32 W flies of each sex, *n* = 16 AR flies of each sex, averaged across 5 days of monitoring) (*** *P* < .001).

difference between females and males during the LP (2-tailed *t*-test = .467, *df* = 27, *P* = .623). The mean locomotor activity level (±SE) of AR virgin males and AR virgin females during the DP was 100.5 ± 10.5 counts and 68.6 ± 9.6 counts respectively. In contrast to the W flies, there was a statistical difference in locomotor activity between the AR female and male flies during the DP (2-tailed *t*-test = 2.236, *df* = 29, *P* = .0329).

**3.1.3. Comparison of locomotor activity between W and AR virgin flies.** There was a significant difference in the mean locomotor activity during the LP between W and AR virgin flies of both sexes (2-tailed *t*-test = 11.398, *df* = 42, *P* < .0001 for males and *t*-test = 10.926, *df* = 43, *P* < .0001 for females) (Fig 4).

## 3.2. Rest episodes

All observed flies preferred to rest near the ventilation end or in the middle of the glass tube. During the LP, resting flies body posture varied, with wings that could be either parallel to the body or slightly extended. Immobile phases could be alternated by self cleaning behaviors (S1 Fig and S1 Video).

During DP, their body was close to the tube surface, with the hind pair of legs extended parallel to the abdomen. Their wings were also parallel to the body. Protrusion and retraction of the proboscis were observed in both sexes, in otherwise immobile flies (S2 Fig and S2 Video).

**3.2.1. Rest episodes of W flies.** During the LP, the mean number of rest episodes (±SE) of W virgin males was 12.1 ± 1.1 with a mean duration (±SE) of 15.7 ± 1.5 min. The mean number of rest episodes (±SE) of W virgin females was 16.8 ± 1.2 with a mean duration (±SE) of 15.5 ± 0.8 min. There was a significant difference between virgin male and female flies in the mean number of rest episodes (2-tailed *t*-test = - 2.697, *df* = 61, *P* = .009), but not in their mean duration (2-tailed *t*-test = .193, *df* = 45, *P* = .847).

During the DP, the mean number of rest episodes (±SE) of W virgin males was 2.9 ± 0.2 with a mean duration (±SE) of 309.0 ± 21.1 min. The mean number of rest episodes (±SE) of W virgin females was 3.8 ± 0.2 with a mean duration (±SE) of 255.8 ± 16.8 min. There was a

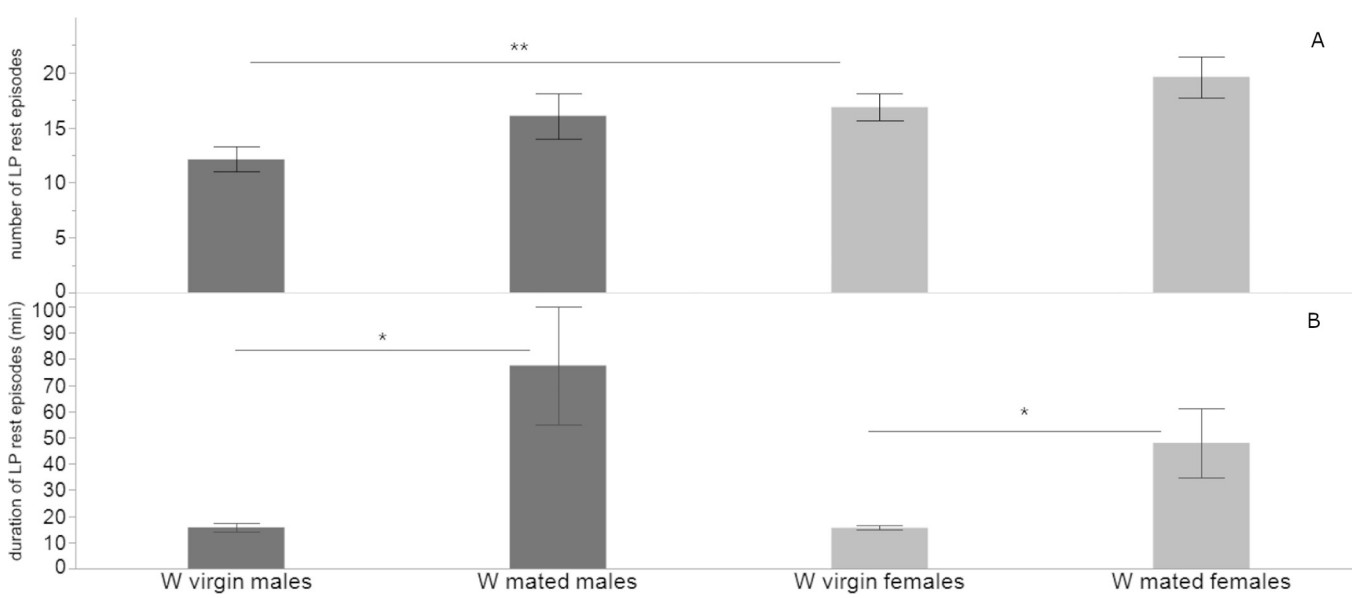

**Fig 5. Rest episodes during the LP of W flies.** Mean (±SE) number (A) and duration (B) of rest episodes of W virgin and mated flies of both sexes during the LP (* $P < .05$, ** $P < .01$).

significant difference between the two sexes in the mean number of rest episodes (2-tailed $t$-test = -2.602, $df$ = 58, $P$ = .011), but not in their mean duration (2-tailed $t$-test = 1.708, $df$ = 58, $P$ = .0902).

During the LP, the mean number of rest episodes (±SE) of W mated males was 16.0 ± 2.0 with a mean duration (±SE) of 77.5 ± 22.3 min. The mean number of rest episodes (±SE) of W mated females was 19.6 ± 1.8 with a mean duration (±SE) of 47.9 ± 13.3 min. There was no significant difference between the two sexes in the mean number of rest episodes (2-tailed $t$-test = -1.261, $df$ = 28, $P$ = .217), nor their mean duration (2-tailed $t$-test = 1.133, $df$ = 23, $P$ = .268).

During the DP, the mean number of rest episodes (±SE) of W mated males was 4.0 ± 0.5 with a mean duration (±SE) of 250.2 ± 39.9 min. The mean number of rest episodes (±SE) of W mated females was 4.3 ± 0.3 with a mean duration (±SE) of 219.2 ± 29.6 min. There was no significant difference between the two sexes in mean number of rest episodes (2-tailed $t$-test = - 0.502, $df$ = 26, $P$ = .619), nor their mean duration (2-tailed $t$-test = .624, $df$ = 26, $P$ = .537).

W mated males differed in the mean duration of rest episodes compared to W virgin males (2-tailed $t$-test = -2.747, $df$ = 14, $P$ = .0156), but no difference was detected in their number during the LP. No difference was detected in the number and mean duration of rest episodes during the DP between the two groups.

Similarly, W virgin and mated females differed in the mean duration of rest episodes during the LP (2-tailed $t$-test = - 2.426, $df$ = 15, $P$ = .0282), but not in their number. No difference was detected in the mean duration and number of rest episodes during the DP (Figs 5 and 6).

**3.2.2. Rest episodes of AR flies.** During the LP, the mean number of rest episodes (±SE) of AR virgin males was 9.7 ± 0.9 with a mean duration (±SE) of 110.6 ± 15.5 min. The mean number of rest episodes (±SE) of AR virgin females was 16.0 ± 1.0 with a mean duration (±SE) of 49.4 ± 5.0 min. There was a significant difference between the two sexes in the mean number of rest episodes (2-tailed $t$-test = - 4.599, $df$ = 29, $P$ < .0001),as well as the mean duration (2-tailed $t$-test = 3.735, $df$ = 18, $P$ = .0015).

During the DP, the mean number of rest episodes (±SE) of AR virgin males was 9.3 ± 0.7 with a mean duration (±SE) of 116.3 ± 15.2 min. The mean number of rest episodes (±SE) of

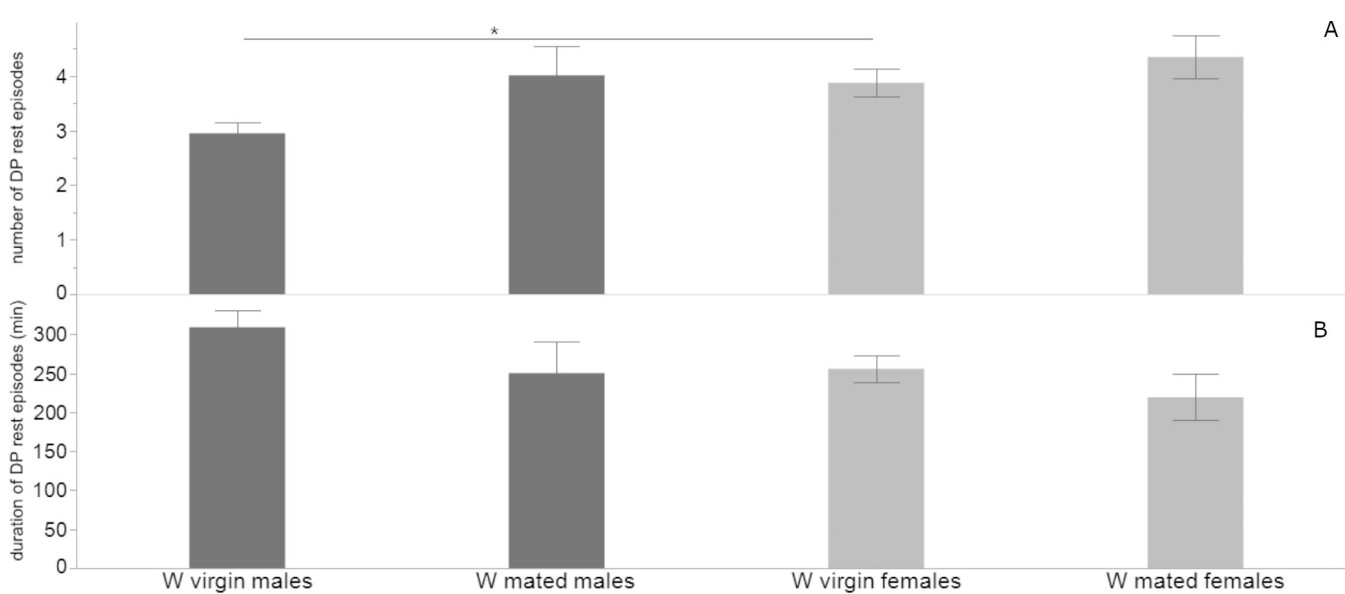

**Fig 6. Rest episodes during the DP of W flies.** Mean (±SE) number (A) and duration (B) of rest episodes of W virgin and mated flies of both sexes during the DP (* *P*< .05).

AR virgin females was 5.6 ± 0.4 with a mean duration (±SE) of 194.1 ± 25.0 min. There was a significant difference between the two sexes in the number of rest episodes (2-tailed *t*-test = 4.243, *df* = 24, *P* = .0003), and the mean duration as well (2-tailed *t*-test = -2.656, *df* = 24, *P* = .0136).

**3.2.3. Comparison of rest episodes between W and AR male flies.** W virgin males and AR virgin males differed in the mean number of rest episodes (2-tailed *t*-test = -8.142, *df* = 17, *P* < .0001) and their mean duration (2-tailed *t*-test = 7.512, *df* = 45, *P* < .0001) during the DP. They also differed in the mean duration of rest episodes (2-tailed *t*-test = -6.866, *df* = 15, *P* < .0001), but not in their number (2-tailed *t*-test = 1.633, *df* = 44, *P* = .109) during the LP. Interestingly, the mean number and duration of rest episodes for AR males, did not differ between the LP and DP (Fig 7).

## 4. Discussion

This study indicates that artificial rearing, mating status and photoperiod phase affect the locomotor activity of adults of the olive fruit fly. W olive fruit flies are mostly active during the LP, and rest episodes during this phase have a mean duration of 15 min. Mate searching and courtship in this species take place during late evening and W virgin males have exhibited an increasing locomotor activity towards the end of the LP (Fig 1), in accordance to past studies [32]. Locomotor activity of W flies during the DP takes mostly during the first hours after lights off. The mean duration of rest episodes during the DP ranges between 255–300 min for both males and females.

W females have lower locomotor levels than males, perhaps due to their heavier body. W mated males have reduced locomotor activity levels and longer rest episodes of compared to W virgin ones. However, W mated females have similar locomotor levels compared to W virgin ones, but experience rest episodes of longer duration. Mated females have been observed trying to oviposit on the glass tube surface (personal observation). This static behavior during which the fly does not change its position inside the tube for several minutes may be recorded as a rest episode by the LAM device. AR flies have lower locomotor activity levels compared to

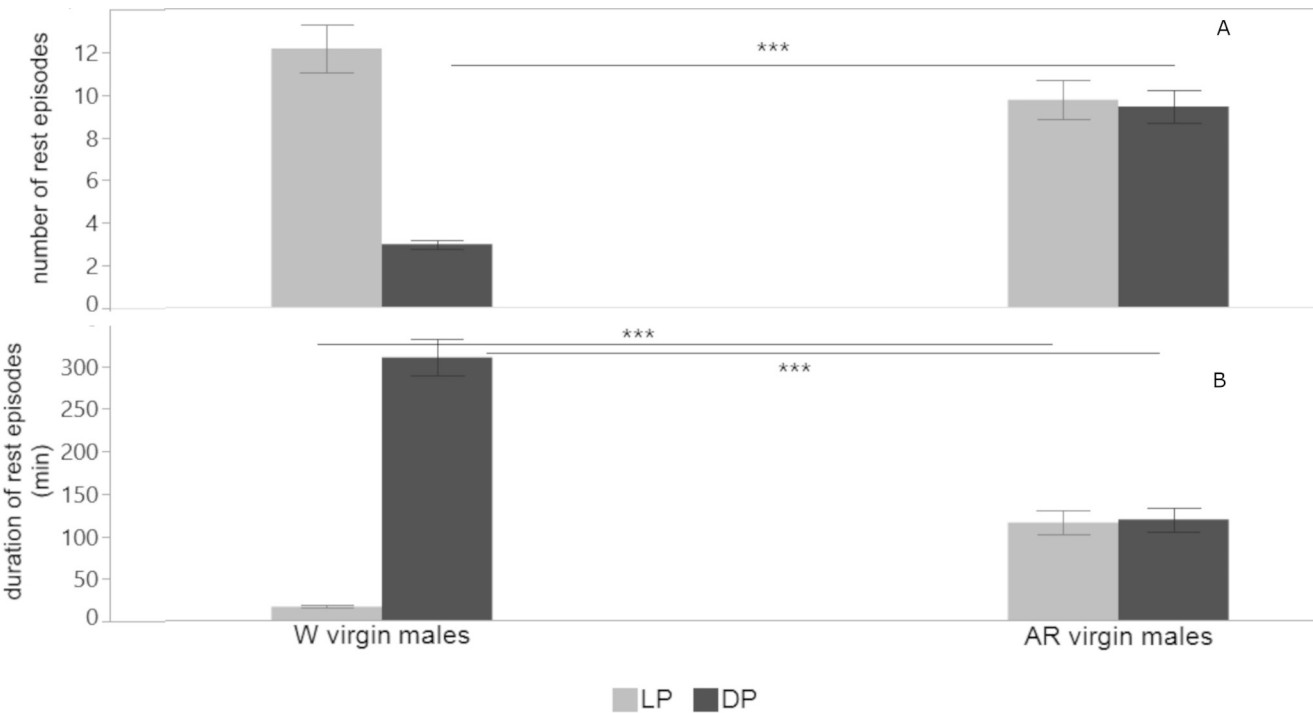

**Fig 7. Comparison of rest episodes between W and AR virgin male flies.** Mean (±SE) number (A) and duration (B) of rest episodes during the LP (light grey color) and during the DP (dark grey color) for W and AR virgin males (*** *P< .001*).

W ones, in accordance to many studies with mass reared insects [33]. The peak of the AR virgin males' locomotor activity is earlier than the W virgin ones (Fig 5), and earlier mating times have been observed in laboratory adapted populations of *B. oleae* [34], *B. tryoni* [35] and *B. cucurbitae* [36]. We also found that during the DP, AR virgin males have a higher number of rest episodes of shorter mean duration compared to W virgin males.

Bertolini et al. [6], compared the locomotor activity between a wild-type and a self-limiting strain of *B. oleae*, the wild-type strain refering to *Argov* and *Democritus* strains that have been laboratory adapted. These strains' activity/h of male flies during the day was found to be 17 counts/h and is similar with our findings of AR mature males' day activity/h (363 counts/14 h = 25.9 counts/h). The reduced activity and high mortality of flies that Bertolini et al. noticed, could be explained by the smaller tubes (10 mm diameter) used to house the flies in the LAM device compared to our larger ones (25 mm diameter), which allowed the olive fruit flies to move more freely and be less stressed.

The differences that have been detected in the locomotor activity levels and rest patterns between W and AR male olive fruit flies can be attributed to laboratory adaptation [37] and could impact the AR flies' survival, dispersion and ability to compete with the wild population. Increased locomotor activity has been previously associated with higher mating success in Tephritidae [38]. Mass reared *B. oleae* and *B. cucurbitae* show a reduced flight ability compared to wild flies [34, 39], which negatively affects their dispersal ability when they are field released [40]. Laboratory adaptation can change the biological traits of weevils [41], affect the fitness of parasitoids used for biological control [42], or lead to loss of stress resistance to *D. melanogaster* [43].

Selective pressures during colonization of *B. oleae* can result in genetic changes that in turn have negative effects on the laboratory reared flies' fitness and competition ability with wild ones [44]. Laboratory-adapted *B. curcubitae* strains that during domestication process were

artificially selected for short larval developmental time and early reproductive age exhibited changes in traits like shorter circadian periods and delayed mating time compared to the wild flies [45].

Our study showed that AR olive fruit flies had more fragmented rest pattern and less total time of night rest compared to W flies. Reduced levels of day activity for AR flies could be explained by the shorter rest episodes during the night. Although in this work we could not define the inactivity episodes during the scotophase as sleep, the relaxed body posture of flies (S2 Fig, S2 and S3 Videos) and the long duration of these inactive episodes (255–300 min) suggest that it is indeed a resting phase. In Drosophila, where inactive episodes have been demonstrated as sleep-like states, it was found that when sleep is disrupted, then it may occur during a period of the day that is normally dedicated to other activities, such as feeding and mating. Also, the loss of sleep can have a negative impact in drosophila fly's brain function [19], although the sleep deprivation effects in adults can be reversed through recovery sleep [46, 47]. In insects, sleep can also impact fitness and affect the reproductive output and development [48].

It is recognized that the adaptation of wild animals to artificial rearing conditions is the cause of altered traits [49]. However, for the successful implementation of SIT it is important to avoid the modification of the insect's behavior which may result in poor field dispersion, reduced sexual fitness or incompatibility with the target insect population in the wild [37]. Bioassays of locomotor activity and LAM devices can be used to estimate the dispersal capacity of laboratory reared flies away from release points [50], an essential attribute of flies for an effective SIT. Also, activity levels provide information about a fly's internal state and improvements in laboratory rearing processes are necessary in order to produce more active flies. Flies that do not exhibit adequate quality traits in optimal laboratory conditions, are unlikely to perform well in the challenging conditions of the field [51].

Future experimental bioassays should be repeated under natural conditions, where environmental stimuli are more abundant and variable compared to the laboratory conditions, can reveal more details about the diurnal rhythms of olive fruit flies.

## Supporting information

**S1 Fig. Resting male (left) and female (right) olive fruit flies during the light period.** Photograph by A. Terzidou.
(TIF)

**S2 Fig. Resting female olive fruit fly during the dark period.** Photograph by A. Terzidou.
(TIF)

**S1 Video. Resting female olive fruit fly during the light period.** Video by A. Terzidou.
(MP4)

**S2 Video. Ventral view of a resting female olive fruit fly during the dark period.** Video by A. Terzidou.
(MP4)

**S3 Video. Lateral view of a resting male olive fruit fly during the dark period.** Video by A. Terzidou.
(MP4)

## Acknowledgments

We would like to thank Eleni Koutsogeorgiou for reading and improving with her suggestions the language of the manuscript.

## Author Contributions

**Conceptualization:** Anastasia M. Terzidou, Dimitrios S. Koveos, Nikos T. Papadopoulos, James R. Carey, Nikos A. Kouloussis.

**Data curation:** Anastasia M. Terzidou.

**Formal analysis:** Anastasia M. Terzidou, Dimitrios S. Koveos, Nikos A. Kouloussis.

**Investigation:** Anastasia M. Terzidou.

**Methodology:** Anastasia M. Terzidou, Dimitrios S. Koveos, Nikos T. Papadopoulos, James R. Carey, Nikos A. Kouloussis.

**Project administration:** Dimitrios S. Koveos, Nikos A. Kouloussis.

**Resources:** Nikos T. Papadopoulos, James R. Carey, Nikos A. Kouloussis.

**Supervision:** Dimitrios S. Koveos, Nikos T. Papadopoulos, James R. Carey, Nikos A. Kouloussis.

**Writing – original draft:** Anastasia M. Terzidou.

**Writing – review & editing:** Anastasia M. Terzidou, Dimitrios S. Koveos, Nikos T. Papadopoulos, James R. Carey, Nikos A. Kouloussis.

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
