## [Decision Letter · Decision Letter 0]

17 Oct 2022

PONE-D-22-24097Laboratory rearing alters activity and sleep patterns in the olive fruit flyPLOS ONE

Dear Dr. Kouloussis,

Thank you for submitting your manuscript to PLOS ONE. After careful consideration, we feel that it has merit but does not fully meet PLOS ONE’s publication criteria as it currently stands. Therefore, we invite you to submit a revised version of the manuscript that addresses the points raised during the review process.

 First, apologies for the long delay to send the reviews due to reviewer prior time commitments. The manuscript is of considerable interest. However the two reviewers have raised some important issues that need to be addressed. Especially reviewer 1 raises significant methodological issues that need to be thoroughly addressed as they will contribute significantly to clarity and ease of interpretation. All other comments and suggestions should also be heeded.

We look forward to receiving your revised manuscript.

Kind regards,

Efthimios M. C. Skoulakis, PhD

Academic Editor

PLOS ONE

Journal Requirements:

https://www.frontiersin.org/articles/10.3389/fnins.2019.01271/full

https://pubmed.ncbi.nlm.nih.gov/11854509/

https://faseb.onlinelibrary.wiley.com/doi/epdf/10.1096/fj.202001107R

In your revision ensure you cite all your sources (including your own works), and quote or rephrase any duplicated text outside the methods section. Further consideration is dependent on these concerns being addressed.

Reviewers' comments:

Reviewer's Responses to Questions

**Comments to the Author**

1. Is the manuscript technically sound, and do the data support the conclusions?

Reviewer #1: No

Reviewer #2: Yes

2. Has the statistical analysis been performed appropriately and rigorously? 

Reviewer #1: Yes

Reviewer #2: Yes

3. Have the authors made all data underlying the findings in their manuscript fully available?

Reviewer #1: Yes

Reviewer #2: No

4. Is the manuscript presented in an intelligible fashion and written in standard English?

Reviewer #1: No

Reviewer #2: Yes

5. Review Comments to the Author

Reviewer #1: The manuscript contains two major problems. First the inactivity is not clearly defined as sleep or what kind of sleep. Fatigue, lethargus, or true sleep? Is this inactivity accompanied with circadain timing, fixed sleep site, fixed sleep posture, rebound from the deprivation of spleep or fixed encephalographic patterns? Second, the change is associated with any circadian parameters like phase, amplitude, period, synchrony etc. Is the change associated with any circadian gene expression dynamics? We cannot make further scientifia analysis from this point. Current condition remains incomplete condition.

Reviewer #2: The manuscript refers to the results of an original research that explores behavioral traits of adults of the olive pest Bactrocera oleae using an automatic system developed for Drosophila. In comparison with similar behavioral studies carried out in the past, the automatic device allows to collect much data, working also in the darkness. The manuscript shows very interesting findings obtained comparing also wild adults to artificially reared ones. These outcomes will improve basic knowledge on insect and animal behavior providing also important insights that should be taken into account for practical applications (to produce high quality mass reared insects for SIT purposes).

Overall, the manuscript is well written, hypotheses and objectives of the research are clearly defined, the methodology is also precisely described. Results are supported by enough data, correctly analyzed (although I think the automatic device might have allowed to perform more replications). The Result chapter should be amended for a recurrent misprint. In the discussion chapter findings are compared with and supported by outcomes from most relevant literature, although some citations were missing.

My main comments and suggestions on the manuscript are reported in the point-to-point attached file.

6. PLOS authors have the option to publish the peer review history of their article (what does this mean?). If published, this will include your full peer review and any attached files.

Reviewer #1: No

Reviewer #2: No

---

## [Author Response · Author response to Decision Letter 0]

1 Dec 2022

Response to the Editor comments:

Journal Requirements:

Response 1. We updated the revised MS according to the style requirements. Figure titles that were not included previously are added. The references style was also adjusted.

• https://www.frontiersin.org/articles/10.3389/fnins.2019.01271/full [48]

Response 2. The phrase has been re-written with an additional reference to support it (lines 355-356 of unmarked MS): In drosophila flies, the loss of sleep can have a negative impact in their brain function [19], although the sleep deprivation effects in adults can be reversed through recovery sleep [46, 47]. 

• https://pubmed.ncbi.nlm.nih.gov/11854509/ [50]

Response 2. The text and reference have been removed, after we re-wrote the discussion section accordingly to reviewers’ suggestions.

• https://faseb.onlinelibrary.wiley.com/doi/epdf/10.1096/fj.202001107R [44] 

Response 2. The text and reference have been removed, after we re-wrote the discussion section accordingly to reviewers’ suggestions.

In your revision ensure you cite all your sources (including your own works), and quote or rephrase any duplicated text outside the methods section. Further consideration is dependent on these concerns being addressed.

Response 3. No field access permit was needed, as infested olive fruits were not collected from a commercial grove, but from abandoned or ornamental trees (line number 85-87 of the unmarked MS)

Response 4. We added the scholarship details received by AMT from the State Scholarship Foundation in the Funding Information section. The correct Financial disclosure is also given in the latest cover letter.

Response 5. We have uploaded the dataset at zenodo repository. https://doi.org/10.5281/zenodo.7221901

Response to Reviewer’s #1 comments:

Reviewer #1: The manuscript contains two major problems. First the inactivity is not clearly defined as sleep or what kind of sleep. Fatigue, lethargus, or true sleep? Is this inactivity accompanied with circadian timing, fixed sleep site, fixed sleep posture, rebound from the deprivation of sleep or fixed encephalographic patterns? Second, the change is associated with any circadian parameters like phase, amplitude, period, synchrony etc. Is the change associated with any circadian gene expression dynamics? We cannot make further scientific analysis from this point. Current condition remains incomplete condition.

Response: We agree that the term sleep has to be defined by the parameters that the reviewer has listed. We have conducted an additional bioassay with wild olive fruit flies to observe the preferred body posture and position in the tubes were the flies are maintained. Photos and videos are provided as supporting information. 

Encephalographic patterns and gene expression research cannot to be conducted in our laboratory. Further bioassays with laboratory reared flies were not possible, as we no longer maintain the specific colony in our laboratory. 

Since we cannot define the episodes of inactivity as sleep episodes, we used the term “rest” throughout the text. The discussion was re-written to support more strongly the importance of quality laboratory reared flies for the SIT, as suggested by the other reviewer. The parts of the discussion referring to the sleep state of flies were removed.

Reviewer #2: The manuscript refers to the results of an original research that explores behavioral traits of adults of the olive pest Bactrocera oleae using an automatic system developed for Drosophila. In comparison with similar behavioral studies carried out in the past, the automatic device allows to collect much data, working also in the darkness. The manuscript shows very interesting findings obtained comparing also wild adults to artificially reared ones. These outcomes will improve basic knowledge on insect and animal behavior providing also important insights that should be taken into account for practical applications (to produce high quality mass reared insects for SIT purposes).

Overall, the manuscript is well written, hypotheses and objectives of the research are clearly defined, the methodology is also precisely described. Results are supported by enough data, correctly analyzed (although I think the automatic device might have allowed to perform more replications). The Result chapter should be amended for a recurrent misprint. In the discussion chapter findings are compared with and supported by outcomes from most relevant literature, although some citations were missing.

My main comments and suggestions on the manuscript are reported in the point-to-point attached file.

Response to reviewer #2

The Result chapter has been amended for a recurrent mistake, as suggested. The discussion section has been re-written, with more comments of the importance of the quality of laboratory reared flies used for SIT, supporting references and also mentioning the previous relevant work of researchers like Economopoulos et al. Point-to-point suggestions were addressed.

---

## [Decision Letter · Decision Letter 1]

19 Jan 2023

PONE-D-22-24097R1Artificial diet alters activity and rest patterns in the olive fruit flyPLOS ONE

Dear Dr. Kouloussis,

Thank you for submitting your manuscript to PLOS ONE. After careful consideration, we feel that it has merit but does not fully meet PLOS ONE’s publication criteria as it currently stands. Therefore, we invite you to submit a revised version of the manuscript that addresses the points raised during the review process.

though the manuscript was improved additional changes as suggested by the reviewer are necessary. Please address all requested changes and suggestions.

We look forward to receiving your revised manuscript.

Kind regards,

Efthimios M. C. Skoulakis, PhD

Academic Editor

PLOS ONE

Journal Requirements:

Reviewers' comments:

Reviewer's Responses to Questions

**Comments to the Author**

1. If the authors have adequately addressed your comments raised in a previous round of review and you feel that this manuscript is now acceptable for publication, you may indicate that here to bypass the “Comments to the Author” section, enter your conflict of interest statement in the “Confidential to Editor” section, and submit your "Accept" recommendation.

Reviewer #2: All comments have been addressed

2. Is the manuscript technically sound, and do the data support the conclusions?

Reviewer #2: Yes

3. Has the statistical analysis been performed appropriately and rigorously? 

Reviewer #2: Yes

4. Have the authors made all data underlying the findings in their manuscript fully available?

Reviewer #2: Yes

5. Is the manuscript presented in an intelligible fashion and written in standard English?

Reviewer #2: Yes

6. Review Comments to the Author

Reviewer #2: The revised manuscript was notably changed and improved, mainly thanks to criticism raised by the reviewer 1. All the issues have been addressed. Moreover, I found the additional experiment useful and new observations and comments explain the results more clearly. According to the new accounts, it seems that flies are not perfectly still during rest episodes since they move their proboscis both in light and dark periods. I understand this is beyond the manuscript’s purpose, however I wonder whether this behavior might be an “involuntary” movement or what else. Was it observed in both sexes? Might it be possibly related to their physiological state?

I am listing some very minor revisions, although I recommend to check the revised manuscript for English language since some sentences do not sound correct.

Line 130: edit “Glycine” instead of “Glysine”

Lines 168-174: specify that wild olive fruit flies have been used.

Line 354: delete “of”

Line 422: I think you intended writing “from” instead of “for”

Line 439: Figure S1: Can you specify in the figure caption if the flies are a male and a female? That cannot be clearly evaluated for the fly on the left.

Line 441: Please, specify in the figure caption “Resting olive fruit fly female during”

Lines 442: Please, specify in the figure caption if the fly is a male or a female

Lines 444: Please, specify in the figure caption if the fly is a male or a female

7. PLOS authors have the option to publish the peer review history of their article (what does this mean?). If published, this will include your full peer review and any attached files.

Reviewer #2: No

---

## [Author Response · Author response to Decision Letter 1]

30 Jan 2023

Response to Reviewer’s #2 comments:

Reviewer #2: The revised manuscript was notably changed and improved, mainly thanks to criticism raised by the reviewer 1. All the issues have been addressed. Moreover, I found the additional experiment useful and new observations and comments explain the results more clearly. 

According to the new accounts, it seems that flies are not perfectly still during rest episodes since they move their proboscis both in light and dark periods. I understand this is beyond the manuscript’s purpose, however I wonder whether this behavior might be an “involuntary” movement or what else. Was it observed in both sexes? Might it be possibly related to their physiological state?

Response: (Line 244 of the Revised Manuscript with track changes) “Protrusion and retraction of the proboscis were observed in both sexes, in otherwise immobile flies.” 

We noted that this behavior was observed in both sexes. It could be an involuntary movement of the fly, but as we lack any more data to support this hypothesis, we did not elaborate any further on this topic in the manuscript.

I am listing some very minor revisions, although I recommend to check the revised manuscript for English language since some sentences do not sound correct.

The manuscript was checked by a proficient colleague, mentioned in the Acknowledgment section.

Responses to point to point comments of the reviewer are listed in the uploaded file "Response to Reviewers.docx"

---

## [Decision Letter · Decision Letter 2]

5 Feb 2023

Artificial diet alters activity and rest patterns in the olive fruit fly

PONE-D-22-24097R2

Dear Dr. Kouloussis,

We’re pleased to inform you that your manuscript has been judged scientifically suitable for publication and will be formally accepted for publication once it meets all outstanding technical requirements.

Kind regards,

Efthimios M. C. Skoulakis, PhD

Academic Editor

PLOS ONE

Additional Editor Comments (optional):

Reviewers' comments:

Reviewer's Responses to Questions

**Comments to the Author**

1. If the authors have adequately addressed your comments raised in a previous round of review and you feel that this manuscript is now acceptable for publication, you may indicate that here to bypass the “Comments to the Author” section, enter your conflict of interest statement in the “Confidential to Editor” section, and submit your "Accept" recommendation.

Reviewer #2: All comments have been addressed

2. Is the manuscript technically sound, and do the data support the conclusions?

Reviewer #2: Yes

3. Has the statistical analysis been performed appropriately and rigorously? 

Reviewer #2: Yes

4. Have the authors made all data underlying the findings in their manuscript fully available?

Reviewer #2: Yes

5. Is the manuscript presented in an intelligible fashion and written in standard English?

Reviewer #2: Yes

6. Review Comments to the Author

Reviewer #2: (No Response)

7. PLOS authors have the option to publish the peer review history of their article (what does this mean?). If published, this will include your full peer review and any attached files.

Reviewer #2: No

---

## [Editor Report · Acceptance letter]

10 Feb 2023

PONE-D-22-24097R2 

Artificial diet alters activity and rest patterns in the olive fruit fly 

Dear Dr. Kouloussis:

I'm pleased to inform you that your manuscript has been deemed suitable for publication in PLOS ONE. Congratulations! Your manuscript is now with our production department. 

Kind regards, 

on behalf of

Dr. Efthimios M. C. Skoulakis 

Academic Editor

PLOS ONE